# Occurrence of Postoperative Delirium and the Use of Different Assessment Tools

**DOI:** 10.3390/geriatrics8010011

**Published:** 2023-01-11

**Authors:** Andrea Kirfel, Diane Jossen, Jan Menzenbach, Andreas Mayr, Maria Wittmann

**Affiliations:** 1Department of Anaesthesiology and Intensive Care Medicine, University Hospital Bonn, Venusberg-Campus 1, 53127 Bonn, Germany; 2Institute for Medical Biometry, Informatics and Epidemiology, University Hospital Bonn, Venusberg-Campus 1, 53127 Bonn, Germany

**Keywords:** postoperative delirium, comparison of postoperative delirium assessments, delirium occurrence postoperatively

## Abstract

(1) Background: Postoperative delirium (POD) poses a high risk of worsening outcomes for patients and is also a burden for hospitals. The leading guidelines recommend standardized POD assessment and prevention. The aim of this subgroup analysis of the PRe-Operative Prediction of Postoperative DElirium by Appropriate SCreening (PROPDESC) trial was to compare different delirium assessments and to analyse the frequency of POD on five postoperative days. (2) Methods: This prospective observational trial enrolled 1097 patients in a university hospital from 2018 until 2019. The following POD assessment tools were used for five consecutive days: Confusion Assessment Method for ICU (CAM-ICU) or Confusion Assessment Method for normal ward (CAM), 4 A’s Test (4AT) and Delirium Observation Screening (DOS) scale. (3) Results: In a 5-day visit interval, most new POD developments occurred on the first and second postoperative day. A clear recommendation for a specific POD assessment tool based on our results cannot be given. (4) Conclusions: According to guidelines, a POD assessment should take place on the first five postoperative days, but of these, the first two are those of highest POD occurrence. The POD assessment tool used should at best include direct patient questioning and aspects of patient observation.

## 1. Introduction

Postoperative delirium (POD) is a behavioural neurological syndrome that is associated with an acute state of confusion with disturbances of attention, consciousness, perception, and/or the thought processes [1,2]. It represents a risk for the patient that should not be underestimated and, at the same time, a major challenge for everyday clinical practice and the health care system.

Various studies have shown that patients who suffered delirium after surgery had impaired general cognitive performance in the postoperative period, compared to patients who did not develop delirium [3,4]. Additionally, the risk of developing dementia increases [5]. However, POD not only raises cognitive but also non-cognitive morbidity, whereas the patients’ quality of life generally decreases [6].

Furthermore, POD increases the risk of complications and a prolonged in-hospital stay while patients’ independence decreases and nursing care requirements rise [7,8]. The aim is therefore to recognize the occurrence of delirium as quickly as possible in order to be able to take measures against it. Thus, patients at risk of POD should be regularly tested for possible delirium in routine clinical practice [6,9]. Based on the limited resources in hospital in terms of time and personnel, the question arises whether a POD test can be limited to a certain period of postoperative days [10,11,12].

In addition, the different POD assessment methods vary in their delirium detection performance and in their applicability in studies in clinical practice [13,14,15]. It has already been proven that targeted POD testing is necessary and that standardized care without a test instrument does not sufficiently detect POD [16,17]. One reason for this is the frequent occurrence of hypoactive delirium, which is often not recognized in everyday clinical practice but misinterpreted as fatigue due to the introverted behaviour of the patients [18,19,20].

The PROPDESC study developed an easily applicable preoperative risk score to detect patients at higher risk for POD [21,22]. The aim of this sub-analysis was to determine on which day a delirium has been identified in most patients and to compare the performance of different delirium tests used in this study.

## 2. Materials and Methods

### 2.1. Study Design and Participants

The PROPDESC trial was a prospective single centre, observational trial at the University Hospital Bonn from September 2018 to October 2019 [21,22]. It was registered in the German Registry for Clinical Studies under the number DRKS00015715. The local Ethics Committee at the Medical Faculty of the Rheinische Friedrich-Wilhelms-University Bonn approved the PROPDESC trial. The trial complied within the principles of the declaration of Helsinki. For all 1097 enrolled patients from several surgery disciplines, written informed consent was obtained.

Inclusion criteria contained patients aged 60 years or older and a planned surgery duration of minimum 60 min. Exclusion criteria were emergency procedures, language barriers and a missing compliance with the study protocol.

### 2.2. Data Collection and Endpoint Definition

The data collection of this prospective trial was conducted pre-, intra and postoperatively. Preoperative data collection took place in the anaesthesia outpatient clinic and on the normal ward. Postoperative visits were carried out in the Intensive Care Unit (ICU), Intermediate Care (IMC) and normal ward. Trained doctoral students visited the enrolled patients on the first five consecutive days after surgery, where appropriate after the end of sedation in ICU and IMC. Sedated patients with Richmond Agitation-Sedation Scale (RASS) <−3 were considered as not assessable and therefore the testing for POD was initiated after exceeding this level of sedation according to Confusion Assessment Method for ICU (CAM-ICU) [23,24]. The aim of the PROPDESC study was to develop a preoperative predictive risk score. Thus, different POD assessments were performed in parallel on all 5-visit days in order not to miss any POD diagnosis.

The Delirium Observation Screening (DOS) Scale was carried out for every patient (in ICU, IMC and on normal ward) [25]. The used DOS version consists of 13 questions about early symptoms of delirium that nurses could observe during regular care. Each item could be rated with 0 (normal) or 1 (abnormal). The cut off value of 3 points or more indicates delirium. Based on nursing questioning by the DOS about abnormalities in the past 24 h, the problem of spot examination at study visits should be avoided.

Beside the DOS on the normal ward, the CAM was used [26]. The CAM detects POD with four diagnostic criteria: acute onset and fluctuating course, inattention, disorganized thinking, and altered level of consciousness. The result of this assessment is positive, if acute onset and fluctuating course as well as inattention apply and if disorganized thinking and/or altered level of consciousness are present. CAM was also assessed during the nurse interview and by conversation with the patient.

In addition to DOS and CAM the 4 A’s Test (4AT) was applied in daily rounds on the normal ward too [27]. The 4AT consists of 4 items. The first item assesses the level of alertness. Item two and three are brief cognitive screening tests and item four assesses acute change or fluctuation in metal status. The result of the 4AT examination provides a score between 0 and 12 points. A score of 0 indicates no delirium. Score results between 1 and 3 are intended to suggest possible cognitive impairment. A value of 4 or above indicates the suggestion of a delirium.

In ICU and IMC, the CAM-ICU was applied in addition to DOS [28]. The CAM-ICU is the further development of CAM, especially for mechanically ventilated patients who cannot verbally articulate. The CAM-ICU result is rated as POD diagnosis if items 1, 2, and 3 or 1, 2 and 4 tested positive.

The visit with the different assessment tools took place on 5 consecutive days. However, individual visits did not take place for various reasons, for example, if a patient did not want to be visited on one day or was discharged. Thus, in the case of individual missing visits, it was determined that the POD assessment was considered complete if at least three visits with conducted testing were available. Patients discharged before the third visit without diagnosed delirium were classified as non-delirious on the assumption that they would not subsequently develop delirium in their usual environment. Furthermore, patients who died before the end of the 5 visits without POD, were removed from the group to be analysed.

### 2.3. Statistical Analysis

The statistical analyses were performed using the statistical programming environment R. Descriptive statistics are presented with numbers and percentages (%) for categorical variables. Mean and standard deviation (sd) are presented for continuous variables. Based on the primary endpoint of POD the patient cohort was divided into the POD- and non-POD group. Differences between these groups were analysed using the Fisher’s exact test for categorical variables to check for independence. Differences between continuous variables were analysed with the non-parametric Wilcoxon rank-sum test.

## 3. Results

### 3.1. Participants

Of the 1097 enrolled patients, 4 (0.4%) refused to participate (see Figure 1). Furthermore, 72 (6.6%) surgeries were cancelled, and so the patients were handled like dropouts. Of the 1021 patients enrolled, 30 (2.7%) patients were excluded for analysis because of too many missing visits (>2 missing visits). Furthermore, 15 (1.4%) patients died without the primary endpoint POD and before the end of the 5-day visit period. Thus, 976 patients were included in this sub analysis of the PROPDESC trial.

### 3.2. Characteristics

The delirium incidence in our study was 23% (229 patients). The gender distribution showed 375 (38%) women and 601 (62%) men (see Table 1). The average age of all cases analysed was 72 (±7) years. Delirious patients were significantly older than non-delirious (73 vs. 72 years, *p* = 0.01). To represent the multimorbidity of the observed patient cohort, the American Society of Anaesthesiology (ASA) Physical Status Classification System was used as a surrogate parameter. Patients who developed POD showed significantly higher ASA levels (level 3 and 4: 85% vs. 56%; *p* < 0.001). Furthermore, 60% (138) of the delirious patients had a cardiac surgical procedure previously.

### 3.3. Visit Distribution and Occurrence of POD

The postoperative visit with the POD test battery was performed on five consecutive days. Based on RASS identification in ICU, a proportion of patients were not tested for POD until the end of sedation and therefore not directly on the first postoperative day. This resulted in a considerable shift from the day of surgery to the postoperative visit interval. In Appendix A, the number of POD assessments on each POD visit day in comparison to the postoperative day are shown. In 91% to 92% of visits, the visit day and the postoperative day corresponded.

The majority of POD determinations were made on the 2nd day. However, this does not mean that most patients became delirious for the first time on the second day, as many patients were delirious for more than one day. The majority of first POD occurrence with 49% took place on visit day 1 (see Table 2).

Table 3 shows that most positive delirium tests occurred in ICU and IMC on visit day one and two. On day one, 80 (71%) positive POD results were generated in ICU. On the third day, POD incidence became higher on normal wards (56%) than on ICU (34%) and IMC (9%).

### 3.4. Comparison of Different POD-Assessments

Different delirium assessments per day were performed in the PROPDESC study to avoid missing a positive POD result. Table 4 shows the frequencies of positive POD ratings per assessment tool per visit day. It becomes apparent that CAM-ICU and DOS generated the most positive POD results on the first visit day. Consistent with this is that most POD diagnoses were also found in ICU and IMC on the first visit day across the cohort. In general, the DOS generated the most positive POD results across all 5 visit days. It should be considered in relation to the fact that the DOS was the only test instrument applied to all wards.

To get a more detailed information about the different test results Table 5 and Figure 2 shows the CAM-ICU and DOS outcomes in IMC and ICU. On the first day, the most POD patients were found with the CAM-ICU (70%). From the second day of the visit-period, this ratio changed. There, the DOS (range from 88% to 100%) found more positive POD diagnoses than the CAM-ICU (42% to 50%). The combination of the two different tests together achieved less positive POD results. Only in 46% of cases were the CAM-ICU and DOS results both positive.

In Table 6 and Figure 3, the POD results of the different POD assessment tools on the normal ward are shown. In the first three visit days, the DOS identified the most POD patients (range from 64% to 71%). On the fourth and fifth day, the 4AT achieved the most positive test results (63% and 68%). The fact that all three tests were positive at the same time was between 13% and 24% for the 5 visit days.

## 4. Discussion

The incidence of delirium in our study cohort was 23%. As shown by several other studies, the patients who developed POD were significantly older and showed significantly higher multimorbidity [6].

The visit day with the most positive POD assessments was the second day. This result is based on the sum of the newly observed POD outcomes and the patients who had already been delirious on the first day. However, most initial POD findings occurred on the first visit day with 49% of the total POD patients. One reason for the increased occurrence on the first day could be the high number of ICU admissions, as well as the proportion of cardiac surgery patients at 28%. As described in the literature, ICU admission and open heart surgery are high risk factors for the development of POD [29,30,31].

A retrospective study describes a 59% POD detection of the patient cohort on the first day [10]. Hamadnalla et al. describes although, that the most POD detection took place in the morning and not in the evening. As the PROPDESC study also conducted the daily delirium visit in the morning, these results are comparable in this respect. However, it is undisputed that delirium testing twice a day increases the success rate of detection. In another retrospective analyses, the study results show 58% of POD detection with the CAM-ICU on the first postoperative day and only 34% on the second day [12]. In an additional study, 52% of delirious patients were POD positive on the first postoperative day and 21% on the second day [11]. In this study also, the CAM-ICU was used as the only test. This probably explains why the POD detection on the first postoperative day was slightly higher in these comparative studies than in PROPDESC. Since our cohort did not only consist of ICU patients.

Besides the studies, which confirm our findings, there is further literature that gives contrary statements on the first occurrence of POD. In a case–control study in 2006, only 5% of delirious patients were detected on the first postoperative day and 40% on the second postoperative day [32]. In this study, elderly patients with emergency and elective surgeries were included. The patient cohort, however, did not include a cardiac surgery discipline and was tested exclusively with CAM. Thus, these results are not directly transferable to the PROPDESC cohort.

In summary of results, it can be concluded that the first postoperative day of ICU patients is the most important for delirium detection. Based on the PROPDESC study design, no delirium testing was performed in the recovery room to avoid confounding with emergence delirium. However, our results show that POD should still be tested in clinical routine at least on the first and second postoperative day. Furthermore, the current guideline for management of delirium in ICU recommend that a delirium assessment should be standardized on demand or at least once per shift (every 8 h) [9]. The current delirium guideline from the European Society of Anaesthesiology and Intensive Care (ESAIC) recommends a standardized postoperative POD screening for five days [6]. The time required for a POD assessment is between 2 and 5 min, depending on the test method and patient condition. Thus, each POD assessment carried out means a total time expenditure for the nurse in the clinical routine.

Compared to the time for the assessment, especially the patients with a hyperactive delirium can cause much additional work for the medical and nursing personnel, sometimes these patients have to stay on intensive care unit, because supervision on the normal ward is not possible [33]. One could carry out a discussion and an economic calculation of the additional time required by standardized tests. In addition, put the time saved through POD reduction in relation to this. Regardless of the time burden on nurses for regular POD detection and the time saved through POD avoidance, the patient’s health and the prevention of further complications and costs remain the primary goal.

Besides standardized postoperative delirium testing, the establishment of an effective POD assessment is important. In the PROPDESC study, several different assessment tools were used in combination with the aim not to miss a positive POD outcome. Our results have shown that the parallel use of different tools only leads to an agreement of the test results in a low percentage. On the normal ward, between 13% and 24% of the three different tests (DOS, CAM and 4AT) were positive at the same time. In ICU and IMC, both tests (DOS and CAM-ICU) applied were positive at the same time in 46% of cases on the first day of the visit. These results suggest that it is useful to commit to one assessment tool in routine clinical practice. Firstly, to save the time resources of nurses and physicians, and secondly, because parallel testing does not seem to have any benefit in POD detection.

Numerous studies on the detection of delirium have used only one assessment tool. The most commonly listed assessment tools in the review by van Velthuijsen et al. are the CAM and the CAM-ICU [15]. Both instruments have already been translated into different languages and validated several times. Thus, CAM-ICU is one of the most frequently used tools in ICU. The 2015 Delirium Guideline requires RASS (part of the CAM-ICU) as the first assessment of patients before they should be assessed for POD depending on outcome of sedation [9]. In our study cohort, the CAM-ICU found 70% of POD in ICU and IMC on the first visit day. In contrast, the DOS identified 61% POD on the first day. On the second visit day, the CAM-ICU found only 46% of delirious patients and the DOS 94%. These results suggest that patients were more delirious on the first postoperative day in the morning when PROPDESC assessment took place. In the following days, the delirium-typical abnormalities appeared over the course of day and thus the nurse’s statements on the 24 h period were more effective. This result is supported by the recommendations of the POD management guideline in ICU, where regular POD testing every 8 h is recommended in routine clinical practice [9]. A possible reason for the different test quality could be the occurrence of the different delirium subtypes [19,20]. Hypoactive delirium is certainly better detected by a direct patient test.

Based on our results, none of the assessments used can be recommended for ICU or IMC practice, despite the significant percentage differences in identification rates on the different visit days. According to the ESAIC POD-Guideline, the most important thing is to test regularly for POD development and to train the applied assessment tool sufficiently [6].

On the normal ward, the DOS showed the highest delirium rate for the first three visiting days (range between 64% and 71%). On the fourth and fifth visit day, the 4AT showed the highest POD rate between 63% and 68%. Here, the occurrence of POD symptoms seems to be different from ICU. On the first three days, the nurse survey (DOS) on patient behaviour in the last 24 h led to the most positive POD outcomes. In contrast, on the following days, at the morning assessment time, most positive POD results were found using the direct patient questionnaire (4AT). The CAM did not achieve the highest POD rate by the nurse survey on any visit day, in contrast to the test results of Wong et al. [34]. However, it showed a much higher POD rate than the 4AT on the first three days. This supports the maintenance that POD symptoms on the first three days seem to have occurred less frequently in the morning on the normal ward. This conclusion is based on the assumption that less delirium was found in the direct patient testing with the 4AT in the morning, but the assessment by CAM was supported by observations of the nursing staff from the previous 24 h.

In a randomized controlled trial in internal medicine patients, the 4AT was tested as superior to CAM in its POD detection rate [35]. In other study results, however, both the CAM and the 4AT are named as reliable test instruments in the normal wards, since both also check inattention and the fluctuating course [14]. These results allow comparisons with our study results on an individual basis. However, the transferability is limited because our presentation focused on the elaboration of the individual postoperative days. Thus, based on our results, it can be summed up that the nursing survey with the DOS performed best in the first three postoperative days and the 4AT performed better on the fourth and fifth day. No POD assessment on the normal wards in our patient cohort achieved a rate of over 71%.

It becomes clear from the test comparisons that the combination of direct patient questioning and the impression of the interviewer or the responsible nurse seems to be the most promising. The CAM combines these two basic ideas, which may explain its wide use in routine testing. However, it has also been shown that POD detection is highly dependent on test performance. Some use it only as a nurse survey, others try it in pure patient contact, and the results show that a combination is very successful [36]. A further developed form is the 3D CAM Assessment, which includes the interviewer’s impression in addition to the direct patient questioning. If the result of this combination is not clear, a nurse or family survey is recommended for clarification [37]. The validation of this newer method provided good test results and seems to be a future-oriented POD assessment model for the normal ward.

Based on the definition of POD outcome in this PROPDESC sub-analysis, there are limitations that make comparability with other study results difficult. The main limitation is that POD results are not based on a gold standard assessment by a psychiatrist or neurologist. According to the PROPDESC study design, the positive POD outcome was determined when at least one assessment of the entire test battery was positive on one of the five visit days. Thus, in this study, sensitivity data per assessment tool were not compiled.

## 5. Conclusions

In a 5-day visit interval, most new POD developments occurred on the first and second postoperative day. However, even on the fifth postoperative day, individual new POD developments were found. A clear recommendation is made for POD testing in the first five postoperative days, according to the ESAIC POD guideline. Furthermore, our study results showed various incongruent frequencies of positive POD test results for the applied assessment tools. A clear recommendation for a specific POD assessment tool based on our results cannot be given. However, the combination of continuous patient observation and direct patient testing has been shown to be the most effective method of recognizing POD.

## Figures and Tables

**Figure 1 geriatrics-08-00011-f001:**
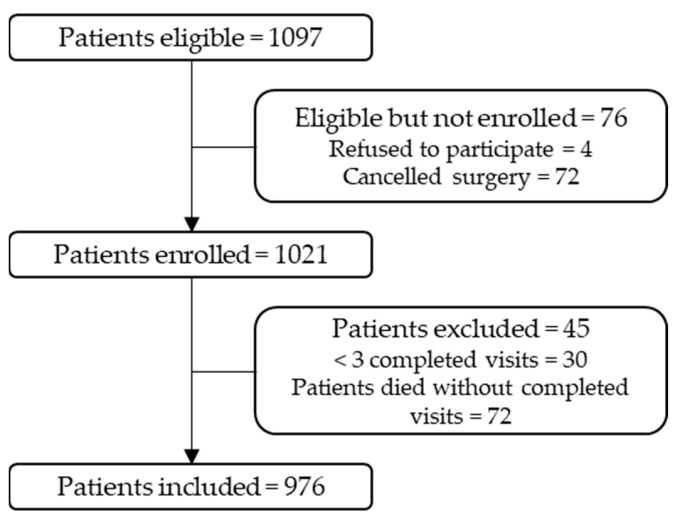
Flow Chart.

**Figure 2 geriatrics-08-00011-f002:**
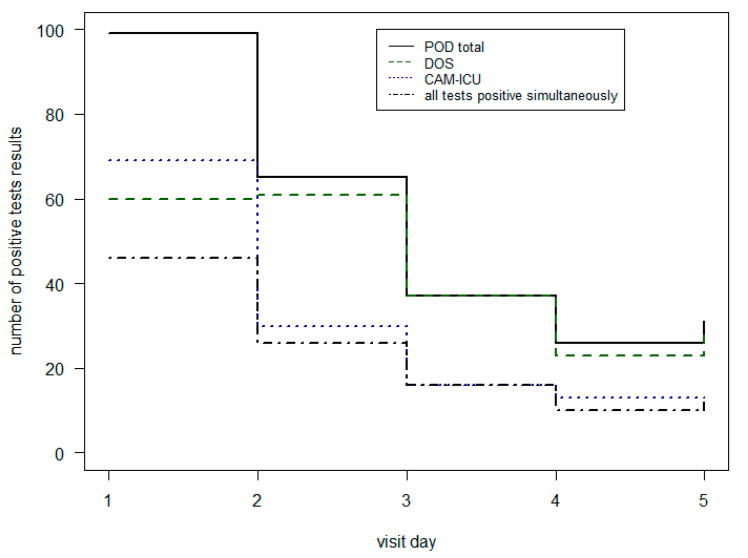
Comparison of positive POD tests in ICU and IMC per visit day.

**Figure 3 geriatrics-08-00011-f003:**
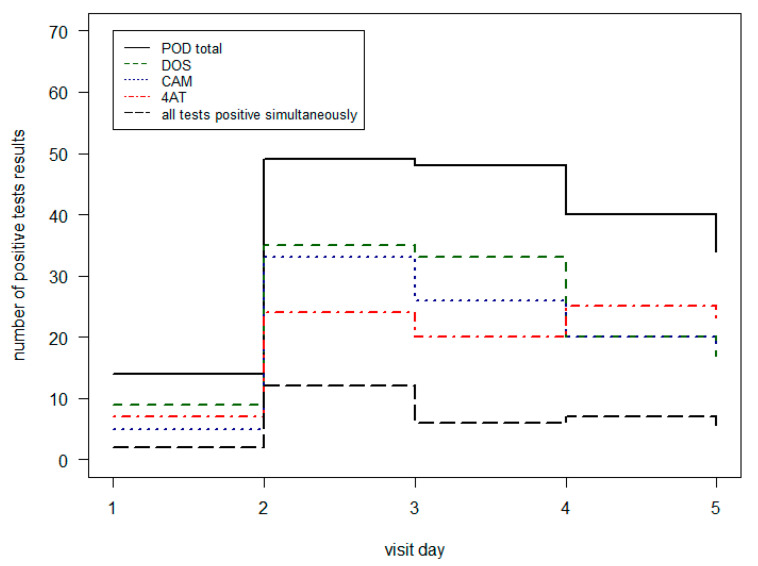
Comparison of positive POD tests on normal ward per visit day.

**Table 1 geriatrics-08-00011-t001:** Characteristics.

Characteristics	Total (*n =* 976)	Non-POD Group (*n =* 747)	POD-Group (*n =* 229)	*p* Value	Missing Values
Age (mean, sd)	72.3 ± 7.3	72.0 ± 7.3	73.3 ± 7.2	0.010	0
Sex				<0.001	0
female	375 (38.4)	311 (41.6)	64 (27.9)		
male	601 (61.6)	436 (58.4)	165 (72.1)		
ASA				<0.001	0
ASA 1	25 (2.6)	21 (2.8)	4 (1.7)		
ASA 2	339 (34.7)	308 (41.2)	31 (13.5)		
ASA 3	544 (55.7)	380 (50.9)	164 (71.6)		
ASA 4	68 (7.0)	38 (5.1)	30 (13.1)		
Surgical discipline				<0.001	0
Others	193 (19.8)	174 (23.3)	19 (8.3)		
Cardiac Surgery	274 (28.1)	136 (18.2)	138 (60.3)		
Orthopaedic Surgery	337 (34.5)	294 (39.4)	43 (18.8)		
Thoracic Surgery	21 (2.2)	17 (2.3)	4 (1.7)		
Abdominal Surgery	123 (12.6)	107 (14.3)	16 (7.0)		
Vascular Surgery	28 (2.9)	19 (2.5)	9 (3.9)		

Data are number (%) unless stated otherwise. POD = Postoperative delirium; ASA = American Society of Anaesthesiology.

**Table 2 geriatrics-08-00011-t002:** Prevalence of first delirium occurrence per visit day.

Visit Day	1	2	3	4	5
POD	113 (49%)	49 (21%)	28 (12%)	23 (10%)	16 (7%)

The percentage refers to the total number of POD patients (229 patients). POD = Postoperative delirium.

**Table 3 geriatrics-08-00011-t003:** Positive POD tests per visit day on the different wards.

Visit Day	1	2	3	4	5
ICU	80 (71%)	58 (51%)	29 (34%)	24 (36%)	27 (42%)
IMC	19 (17%)	7 (6%)	8 (9%)	2 (3%)	4 (6%)
Norm	14 (12%)	49 (43%)	48 (56%)	40 (61%)	34 (52%)

ICU = Intensive Care Unit; IMC = Intermediate Care; Norm = Normal ward.

**Table 4 geriatrics-08-00011-t004:** Positive POD tests per visit day.

Visit Day	CAM-ICU	DOS	CAM	4AT
1	69	85	5	7
2	30	96	33	24
3	16	70	26	19
4	13	48	20	20
5	13	53	18	17

DOS = Delirium Observation Screening; CAM = Confusion Assessment Method; ICU = Intensive Care Unit; 4AT = 4 A’s Test.

**Table 5 geriatrics-08-00011-t005:** Comparison of positive POD tests in ICU and IMC.

Visit Day	POD	CAM-ICU Pos	DOS Pos	CAM-ICU + DOS Pos
1	99	69 (70%)	60 (61%)	46 (46%)
2	65	30 (46%)	61 (94%)	26 (40%)
3	37	16 (43%)	37 (100%)	16 (43%)
4	26	13 (50%)	23 (88%)	10 (38%)
5	31	13 (42%)	30 (97%)	12 (39%)

ICU = Intensive Care Unit; IMC = Intermediate Care; POD = Postoperative delirium; CAM = Confusion Assessment Method; DOS = Delirium Observation Screening; pos = positive.

**Table 6 geriatrics-08-00011-t006:** Comparison of positive POD tests on normal ward.

Visit Day	POD	CAM Pos	DOS Pos	4AT Pos	CAM + DOS + 4AT Pos
1	14	5 (5%)	9 (64%)	7 (50%)	2 (14%)
2	49	33 (67%)	35 (71%)	24 (49%)	12 (24%)
3	48	26 (54%)	33 (69%)	20 (42%)	6 (13%)
4	40	20 (50%)	20 (50%)	25 (63%)	7 (18%)
5	34	18 (53%)	17 (50%)	23 (68%)	5 (15%)

POD = Postoperative delirium; CAM = Confusion Assessment Method; DOS = Delirium Observation Screening; 4AT = 4 A’s Test; pos = positive.

## Data Availability

The data sets generated and analysed during the study are available from the corresponding author on reasonable request. The R code used for the analysis is available from the corresponding author on reasonable request.

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
