# Peer review of "Occurrence of Postoperative Delirium and the Use of Different Assessment Tools"

_geriatrics, 2023, doi:10.3390/geriatrics8010011_

Round 1
Reviewer 1 Report
This manuscript describes a solid study on what assessment to use the best for detecting a delirium after surgery. The findings are interesting especially the change over days in which instrument can detect / measure the delirium best. This issue earns more investigation: what were the differences in symptoms/measures of instruments that made this shift?
It is important to detect a delirium soon and fast after the surgery. This information adds to the knowledge on this topic.
However, different kind surgeries were included in the study. Most of them were cardiac surgeries, in which the most cases of delirium were found.
It would be interesting to investigate whether the findings on ‘best instrument’ to detect POD are the same for the different surgeries or are there differences.
In conclusion, the manuscript describes an important topic and a well performed investigation, yet the analyses and explanations / exploration of the results are a bit shallow.
In addition, I do not understand Appendix A. What is the difference between visit day and post-surgery day?
Author Response
Dear Reviewer 1,
Thank you for this positive feedback.
We agree with you that a quick and easy method to detect postoperative delirium (POD) is of utmost importance. It is true that we included all types of surgery (with an estimated duration of at least 60 minutes). We wanted to get an overview of all our patients, not just those in one surgical specialty. Cardiac surgery patients had the highest incidence of POD at 50%. We were able to compare the cardiac surgery patients who tested positive for POD (n=138) with the non-cardiac surgery patients (n=91) to analyse whether there were differences in the detection of delirium.
Unfortunately, the statistician is on Christmas holiday, so we will not be able to perform these analyses until January. Deadline for revision of the manuscript is tomorrow. If you think this research is important for the manuscript, we are happy to perform these analyses.
Appendix A: Some patients were sedated after surgery, so not all patients could be examined on the first day after surgery. We did not want to exclude these patients because the postoperative stay in the ICU may increase the risk of POD. Therefore, we started testing patients on the first day after sedation. In the table you can see that in 43 patients the first test (day of visit) was done on the second postoperative day and in 15 patients on the third postoperative day.
This is described in lines 74 to 79: "Trained doctoral students visited the enrolled patients on the first five consecutive days after surgery, where appropriate after the end of sedation in ICU and IMC. Sedated patients with Richmond Agitation-Sedation Scale (RASS) < -3 were considered as not assessable and therefore the testing for POD was initiated after exceeding this level of sedation according to Confusion Assessment Method for ICU (CAM-ICU) [23,24]."
Reviewer 2 Report
Interesting work, congratulations. It would have been interesting to have a reference rater to allow for the calculation of sensivity, PPV etc. That would offer the opportunity to compare tests more in detail and give the reader the chance to make a selection as to which test may serve his/ her clinic best for delirium detection. What this paper tells us: distinct test just do not measure the same.
Author Response
Dear reviewer,
thank you for this positive feedback on our manuscript.
Yes, I agree that a reference rater would be great, but according to current opinion, the psychiatric consultation is the gold standard. Unfortunately, with more than 1000 patients included, this could not be implemented within the framework of this study.
Reviewer 3 Report
Excellent work!
Line 127 replace with -because of too many visits instead of “too many visits”.
Author Response
Dear Reviewer,
thank you for this positive feedback on our manuscript!
We changed in line 127 "to" into "too".
Thanks